



# Kinetics of the reactions of OH with CO, NO, NO$_2$ and of HO$_2$ with NO$_2$ in air at 1 atm pressure, room temperature and tropospheric water vapour concentrations

Michael Rolletter[1], Andreas Hofzumahaus[1], Anna Novelli[1], Andreas Wahner[1], and Hendrik Fuchs[1,2]

[1]Institute of Climate and Energy Systems, ICE-3: Troposphere, Forschungszentrum Jülich GmbH, Jülich, Germany
[2]Department of Physics, University of Cologne, Cologne, Germany

**Correspondence:** Hendrik Fuchs (h.fuchs@fz-juelich.de) and Andreas Hofzumahaus (a.hofzumahaus@fz-juelich.de)

**Abstract.** The termolecular reactions of hydroxyl radicals (OH) with carbon monoxide (CO), nitric oxide (NO) and nitrogen dioxides (NO$_2$) and the termolecular reaction of hydroperoxyl radicals (HO$_2$) with NO$_2$ greatly impact the atmospheric oxidation efficiency. Few studies have directly measured the pressure dependent rate coefficients in air at 1 atm pressure and water vapour as third collision partners. In this work, rate coefficients were measured with a high accuracy ($< 5\,\%$) at 1 atm

pressure, room temperature and in humidified air using laser flash photolysis and detection of the radical decay by laser-induced fluorescence. The rate coefficients derived in dry air are: $(2.39 \pm 0.11) \cdot 10^{-13}\,\mathrm{cm^3 s^{-1}}$ for the OH reaction with CO, $(7.3 \pm 0.4) \cdot 10^{-12}\,\mathrm{cm^3 s^{-1}}$ for the OH reaction with NO, $(1.23 \pm 0.04) \cdot 10^{-11}\,\mathrm{cm^3 s^{-1}}$ for the OH reaction with NO$_2$, and $(1.56 \pm 0.05) \cdot 10^{-12}\,\mathrm{cm^3 s^{-1}}$ for the HO$_2$ reaction with NO$_2$. For the OH reactions with CO and NO, no dependence on water vapour was observed for the range of water partial pressures tested (3 to 22 hPa), and for NO$_2$, only a weak increase of 3 %

was measured in agreement with the study by Amedro et al. (2020). For the rate coefficient of HO$_2$ with NO$_2$ an enhancement of up to 25 % was observed. This can be explained by a faster rate coefficient of the reaction of the HO$_2$-water complex with NO$_2$ having a value of $(3.4 \pm 1.1) \cdot 10^{-12}\,\mathrm{cm^3 s^{-1}}$.

## 1   Introduction

The inorganic pressure-dependent reactions of the OH with CO, NO, and NO$_2$ and of HO$_2$ with NO$_2$ link the chemistry of

HOx (the sum of OH and HO$_2$) and NOx (NO and NO$_2$) in the atmosphere and affect largely the chemical transformation of pollutants (Newsome and Evans, 2017). The OH radical is the most important oxidant, reacting with most volatile compounds. Its reaction with pollutants initiates radical chain reactions, in which HO$_2$) radicals are often formed and in which OH can be eventually regenerated. In the troposphere, for example, CO, which is emitted from combustion processes, is oxidised in the termolecular reaction with OH:

$$\mathrm{OH + CO} \;\; \rightleftharpoons \;\; \mathrm{HOCO^*} \xrightarrow{\mathrm{M}} \mathrm{HOCO} \tag{R1}$$

$$\mathrm{HOCO^*} \;\; \rightarrow \;\; \mathrm{H + CO_2} \tag{R2}$$



M is a third body collision partner. The reaction of OH with CO has been studied experimentally and theoretically over a wide range of temperatures and pressures because of its general importance in the planetary atmospheres of Earth and Mars (Atkinson et al., 2004; Burkholder et al., 2020).

Nitrogen oxides are mainly emitted by combustion processes and produced in the atmosphere by lightning. They play an important role in atmospheric radical chemistry in several ways. The reactions of NO with peroxy radicals are responsible for the regeneration of OH radicals. Conversely, the reactions of OH with NO and $NO_2$ and of $HO_2$ with $NO_2$ form products that terminate the cyclic chain reactions between OH and $HO_2$ and can produce long-lived compounds that can act as radical reservoirs. In addition, the oxidation reaction of NO to $NO_2$ by peroxy radicals followed by $NO_2$ photolysis is the only relevant
chemical source of tropospheric ozone (Ehhalt, 1999).

The termolecular reaction of OH with NO produces nitrous acid (HONO):

$$OH + NO + M \rightarrow HONO + M \tag{R3}$$

HONO can be rapidly photolysed so that OH, NO and HONO concentrations are in a photochemical equilibrium at daytime (Kleffmann et al., 2005).

The reaction of OH with $NO_2$ is a termolecular reaction leading to the formation of nitric acid ($HNO_3$) or pernitrous acid (HOONO):

$$\begin{aligned} OH + NO_2 + M \quad &\rightarrow \quad HNO_3 + M \tag{R4} \\ &\rightleftharpoons \quad HOONO + M \tag{R5} \end{aligned}$$

In the lower troposphere, $HNO_3$ is mainly lost by surface deposition due to its long chemical lifetime. The reaction channel
leading to its formation is therefore a net loss of OH radicals and nitrogen oxides. In contrast, HOONO is thermally unstable and decomposes mainly in the boundary layer at mid-latitude temperatures so that there is no net loss of the reactants. If HOONO underwent other atmospheric reactions, its formation would be a radical and $NO_2$ sink, but such reactions have not been reported. The branching ratio between Reaction R4 and Reaction R5 increases with pressure and is approximately 14 % at atmospheric pressure and room temperature (Mollner et al., 2010).

Pernitric acid ($HO_2NO_2$), formed by the termolecular reaction of hydroperoxyl radicals ($HO_2$) and $NO_2$, can decompose thermally in the troposphere so that their concentrations are in a thermal equilibrium (Gierczak et al., 2005):

$$HO_2 + NO_2 + M \rightleftharpoons HO_2NO_2 + M \tag{R6}$$

In the upper troposphere and lower stratosphere, where $HO_2NO_2$ is thermally stable due to the cold temperatures, its subsequent reaction with OH is an important sink for $HO_x$ ($=HO_2+OH$) radicals (Kim et al., 2007). Measurements of $HO_2NO_2$
can also be used to diagnose $HO_2$ and $NO_2$ concentrations, but accurate rate coefficients are required to calculate steady state concentrations.

The reactions of OH with CO, NO and $NO_2$ and the reaction of $HO_2$ with $NO_2$ are termolecular reactions, in which an activated association complex is formed. The rates of dissociation and collisional stabilisation of the activated complex





determine the rate coefficients of the overall reaction. Therefore, the rate coefficients are pressure dependent (expressed as
the number density concentration of the bath gas molecules, $M$), which can be parameterised by the Troe formalism (Troe,
1983). The Troe expression parametrises the rate using high-pressure ($k_\infty$) and low-pressure ($k_0$) limiting rate coefficients. A
"fall-off" transition is described by the broadening factor $F$. The expression used e.g. by IUPAC is (Atkinson et al., 2004):

$$k(M,T) = \frac{k_0 \left(\frac{T}{300\text{K}}\right)^{-m} M \, k_\infty \left(\frac{T}{300\text{K}}\right)^{-n}}{k_0 \left(\frac{T}{300\text{K}}\right)^{-m} M + k_\infty \left(\frac{T}{300\text{K}}\right)^{-n}} \, F \tag{1}$$

where $m$ and $n$ are dimensionless temperature exponents. The broadening factor $F$ is:

$$\log F = \frac{\log F_c}{1 + \left[\log \left(\frac{k_0 \left(\frac{T}{300\text{K}}\right)^{-m} M}{k_\infty \left(\frac{T}{300\text{K}}\right)^{-n}}\right) / N\right]^2} \tag{2}$$

with $N = 0.75 - 1.27 \cdot \log F_c$ and $F_c$ being the broadening factor at the centre of the fall-off transition. The parameterisation
by NASA-JPL is only slightly different.

    Despite the importance of these reactions for the atmospheric cycle of radicals and nitrogen oxides, there are only few studies
that have directly measured their rate coefficients in air at $1\,\text{atm}$ pressure.

The evaluations of rate coefficients are based on the limited data reported in the literature, resulting in notable differences in
the values recommended by NASA-JPL (Burkholder et al., 2020) and IUPAC (Atkinson et al., 2004) at $1\,\text{atm}$. For example, the
recommendations differ by a factor of 1.3 for the OH reaction with NO and by a factor of 1.8 for the $HO_2$ reaction with $NO_2$.
Consequently, the predictions of atmospheric chemistry models that rely on recommendations in databases may be subject to
considerable uncertainties, emphasising the need for further laboratory studies to reduce the uncertainties (Burkholder et al.,
2017; Fiore et al., 2024; Ervens et al., 2024).

    Previous studies have shown that the presence of water vapour can affect the rate coefficients of OH and $HO_2$ reactions
through the formation of a hydrogen-bonded complex between $HO_2$ and a water molecule (Cox and Burrows, 1979; Aloisio
et al., 2000; Kanno et al., 2005; Buszek et al., 2011) or by collisional stabilisation of the activated association complex by
water molecules (Amedro et al., 2020; Sun et al., 2022).

For example, significantly increased rate coefficients have been observed in the self-reaction of $HO_2$ (e.g. Lii et al., 1981;
Kircher and Sander, 1984) or in the reaction of $HO_2$ with $NO_2$ (Sander and Peterson, 1984) at low pressure in the presence of
water vapour. However, with a few exceptions, such as the self-reaction of $HO_2$, possible water vapour dependencies have not
been considered in the NASA-JPL and IUPAC recommendations due to the lack of sufficient experimental data.

    In this work, a laser flash photolysis/ laser-induced fluorescence (LP-LIF) method was used to generate OH radicals by
ozone photolysis in a flow tube and to observe the rate of their chemical decay. Unlike in many pump-and-probe instruments,
the radical detection does not take place in the reaction volume, but in a low-pressure cell, which allows an extremely sensitive
OH fluorescence detection.

    The instrument was originally developed to measure the chemical OH lifetime in ambient air at tropospheric conditions
(Hofzumahaus et al., 2009; Lou et al., 2010). In atmospheric studies, the measured OH lifetime is a valuable kinetic parameter,
which can be used to determine the production and destruction rates of atmospheric OH allowing the quantification of poten-



tially unknown sources and sinks (Hofzumahaus et al., 2009; Lou et al., 2010; Kovacs and Brune, 2001; Martinez et al., 2003; Sadanaga et al., 2004a; Whalley et al., 2011; Fuchs et al., 2013, 2014; Griffith et al., 2016; Yang et al., 2016).

The inverse atmospheric OH lifetime is called the total OH reactivity ($k_{\mathrm{OH}}$) and is equal to the pseudo-first order loss rate coefficient. Its value depends on the concentrations of all atmospheric reactants $i$ (e.g. CO, NOx, hydrocarbons) and their

second order rate coefficients ($k_{\mathrm{OH}+i}$).

$$k_{\mathrm{OH}} = \sum_i k_{\mathrm{OH}+i}[i] \tag{3}$$

In the lower troposphere, observed OH reactivity values are in the range from $1\,\mathrm{s}^{-1}$ to $100\,\mathrm{s}^{-1}$ for conditions ranging from very clean to extremely polluted air (Hofzumahaus et al., 2009; Yang et al., 2016).

In this work, the instrument was used to determine the rate coefficients of the reaction of OH with CO, NO and $NO_2$ and

of $HO_2$ with $NO_2$ in air at atmospheric pressure and room temperature and in the presence of water vapour. Similar reactivity instruments have been used previously for kinetic studies of OH (Sadanaga et al., 2004b; Amedro et al., 2012; Nakashima et al., 2012; Stone et al., 2016; Speak et al., 2020; Berg et al., 2024). The method can also be used to study the kinetics of $HO_2$ radicals by adding excess CO in the flow tube to convert all initially produced OH to $HO_2$ (Nehr et al., 2011, 2012; Zhou et al., 2019). In this work, the reaction of $HO_2$ with $NO_2$ was studied using this approach.

## 2    Methods

### 2.1    Measurement of pseudo-first order rate coefficients

The central components of the laser flash photolysis / laser-induced fluorescence (LP-LIF) instrument used in this work to determine OH and $HO_2$ rate coefficients are a laminar flow tube reactor, in which OH radicals are produced by flash photolysis and an attached fluorescence detection cell for measuring the OH decay (Fig. 1). The flow tube has a total length of $80\,\mathrm{cm}$

and an internal diameter of $40\,\mathrm{mm}$. It is made of black anodised aluminium and is sealed at both ends by fused silica quartz windows with an antireflective coating for $266\,\mathrm{nm}$ (Laser Optics). The distance between the entrance of the flow tube and the sampling point of the detection cell is $50\,\mathrm{cm}$. The air is replaced every $1.8\,\mathrm{s}$ using a mass flow controller (Bronkhorst, Low $\Delta p$ series, flow rate: $21\,\mathrm{l/min}$) backed by a vacuum pump (Vacuubrand, MD4C). For the experiments in this work, the flow tube was kept at room temperature and ambient pressure. Sensors monitor the pressure (Honeywell, PPT), the temperature and

the relative humidity (Vaisala, Humicap) of the gas at the outlet of the flow tube. The flow in the flow tube is laminar with a Reynolds number of 710.

Laser flash photolysis of added ozone is used to generate excited oxygen atoms ($O(^1D)$), which react with the water molecules to form OH on a time scale of nanoseconds in the flow tube:

$$O_3 + h\nu \quad \rightarrow \quad O(^1D) + O_2$$

$$O(^1D) + H_2O \quad \rightarrow \quad 2\,OH \tag{R7}$$





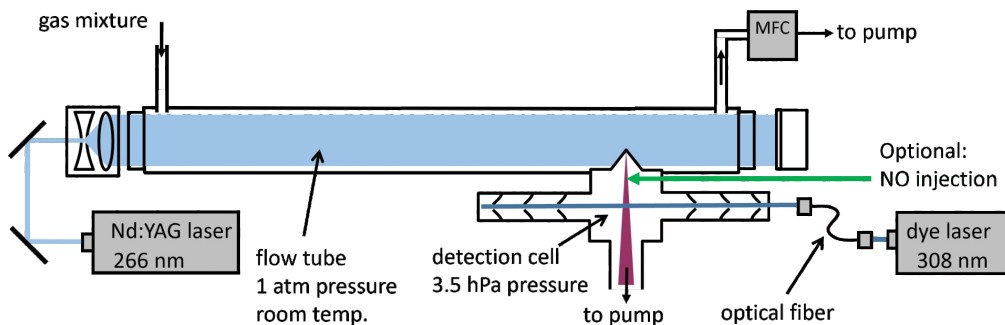

**Figure 1.** Schematic diagram of the instrument. The air mixture in the flow tube is exposed to laser pulses at 266 nm to generate OH radicals by flash photolysis of ozone. Air is sampled through the flow tube using a mass flow controller (MFC) backed by a pump. The decay of OH is measured by laser-induced fluorescence in a low pressure detection cell. Optional injection of NO into the low pressure detection cell allows the detection of $HO_2$ radicals after their chemical conversion to OH.

A frequency-quadrupled Nd:YAG laser (Quantel, Ultra) delivers short pulses (10 ns) of 266 nm radiation at a repetition rate of 1 Hz and pulse energies of 20 to 27 mJ. The laser beam is expanded by an optical telescope to a diameter of 30 mm. Since the collimated photolysis laser beam is not attenuated as it passes through the reaction volume (the optical density is less than $10^{-3}$), the same initial OH concentration is obtained along the axis of the flow tube. Depending on the water vapour concentration and the laser pulse energy, the mean initial OH number concentrations range from $2 \cdot 10^9 \, \mathrm{cm}^{-3}$ to $9 \cdot 10^9 \, \mathrm{cm}^{-3}$. To produce $HO_2$ radicals, 80 ppmv CO can be added to the gas in the flow tube for a rapid conversion of the initially produced OH. The time constant of the conversion is 2 ms.

Once formed, the OH or $HO_2$ radicals react with the reactive components in the air mixture. The concentration decreases following a pseudo first order kinetics for all experimental conditions in this study:

$$[\mathrm{OH}](t) \;=\; [\mathrm{OH}]_0 \exp(-k't) \tag{4}$$

$$[\mathrm{HO_2}](t) \;=\; [\mathrm{HO_2}]_0 \exp(-k't) \tag{5}$$

where $[\mathrm{OH}]_0$ and $[\mathrm{HO_2}]_0$ are the initial radical concentrations and $k'$ is the first order rate coefficient of the exponential decay, which is the sum of the first order rate coefficients of the loss in the reaction with the gaseous reactant and the wall loss.

The time-resolved decay of the radical concentration is measured in a low-pressure detection cell (3.5 hPa), which continuously draws gas from the reaction volume through a conical nozzle (Beam Dynamics, nickel, 0.6 mm orifice, 3.6 l/min flow rate, Fig. 1). For the experiments with $NO_2$, a gold-plated nozzle was used to prevent corrosion of the inlet.

In the detection cell, the OH is excited by pulsed laser radiation at a wavelength of 308 nm matching the rotational absorption line $Q_1(3)$ of the OH($A^2\Sigma, \nu' = 0 \leftarrow X^2\Pi, \nu'' = 0$) band transition. The UV radiation is generated by a custom-built, tunable, frequency-doubled dye laser (Strotkamp et al., 2013), which is pumped by a pulsed frequency-doubled Nd:YAG laser (Spectraphysics, Navigator). The laser pulse repetition rate is 8.5 kHz and the typical UV output power is 20 mW.





This system is also capable of detecting $HO_2$ if the radical is chemically converted to OH before passing through the 308 nm probing laser beam (Fig. 1). This is achieved by injecting pure NO (Air Liquide, purity 99.5 %, flow rate 5 $cm^3$/min at standard conditions) into the sampled gas flow in the detection cell (Nehr et al., 2011, 2012; Miyazaki et al., 2013):

$$HO_2 + NO \rightarrow OH + NO_2 \tag{R8}$$

The added NO was purified by passing it through a cartridge filled with sodium-hydroxide coated silica (Sigma-Aldrich, Ascarite) to avoid spurious OH background signals from the 308 nm photolysis of NO impurities. This method gives almost the same detection sensitivity for $HO_2$ as for OH (Fuchs et al., 2011).

The OH fluorescence is recorded by a multi-channel scaler photon counting system (Becker & Hickl, PMS-400A) with a time resolution of 1 ms over a time period of 1 s. In this instrument, the reaction time is determined by the electronic clock of
the multi-channel scaler, in contrast to flow tube experiments with sliding injectors, where the reaction time is determined from the flow rate of the gas in the reaction volume. The radical detection method using OH fluorescence is extremely sensitive and allows the detection of radical number concentrations in the order of $10^6$ cm$^{-3}$ in 1 atm of air with a measurement time of 1 min (Hofzumahaus et al., 2009; Lou et al., 2010).

The separation of the detection in a low pressure cell from the high pressure reaction volume has several advantages for the
study of OH and $HO_2$ reactions at tropospheric pressures:

– The low pressure OH detection minimises the loss of sensitivity due to quenching of the OH fluorescence, which is particularly efficient for water and $O_2$ (relative rate coefficients in units of $10^{-11}$ cm$^3$s$^{-1}$ at 298 K: $k_{coll}(H_2O) : k_{coll}(O_2) : k_{coll}(N_2) : k_{coll}(Ar) = 6.6 : 1.4 : 0.31 : 0.00036$, Heard and Henderson (2000)). In contrast, previous fluorescence-based studies have often used Ar or $N_2$ as a buffer gas to reduce fluorescence quenching.

– The high detection sensitivity by the fluorescence method allows the use of low initial radical concentrations (a few $10^9$ cm$^{-3}$) and makes thereby the influence of interfering radical-radical reactions and subsequent reactions with products negligible on the time scale of the measured decays.

– The method can be used to determine rate coefficients under typical tropospheric conditions of pressure, temperature and concentrations of water vapour and reactants. This is of particular interest for the study of termolecular reactions, whose
rate coefficients depend on the pressure and the properties of the bath gas molecules.

– Another advantage is the minimal radical wall loss due to the slow diffusion at atmospheric pressure. This allows for the reactions to be studied on a timescale of one second, which is comparable to the typical timescales of HOx reactions in the lower troposphere. The timescale and radical concentrations employed allow experiments to be carried out under pseudo first order conditions with lower reactant concentrations than those used in previous studies. Consequently, the
impact of potentially interfering reactions of the reactants (e.g. the reaction of NO or $NO_2$ with ozone, or the formation of dinitrogen tetroxide ($N_2O_4$) from the self-reaction of $NO_2$) is suppressed. However, unimolecular reactions, such as the reaction of an association product, may become important.



**Table 1.** Mixing ratios of the reactants in the gas mixtures as specified by the suppliers and measured in this work. All reactants were mixed in $N_2$.

| gas cylinder | reactant | supplier | mix. ratio / ppmv (supplier spec.) | mix. ratio / ppmv (measured) | impurities / ppmv (measured) |
|---|---|---|---|---|---|
| A | CO | Linde | $500 \pm 10$ | $500 \pm 10$ | – |
| B | NO | Air Liquide | $9.96 \pm 0.20$ | $9.9 \pm 0.5$ | – |
| C | NO | Air Liquide | $96.3 \pm 1.9$ | $101 \pm 3$ | $(1 \pm 3)\,NO_2$ |
| D | $NO_2$ | Praxair | $520 \pm 10$ | $524 \pm 3$ | $(2 \pm 3)\,NO$ |

## 2.2 Gas mixtures

The gas mixtures overflowing the instrument inlet were prepared in two steps: First by combining flows of dry synthetic air (flow rate $23\,l/min$), humidified synthetic air ($3\,l/min$) and air containing ozone ($0.1\,l/min$). From the combined mixed flow, $1\,l/min$ was continuously sampled by a hygrometer (Vaisala, Humicap) and an ozone analyser (Environment SA, O341M). The remaining flow was combined with a small flow ($< 2\,l/min$) of a reactant gas premixed in $N_2$. All gas flows were controlled by mass flow controllers (Bronkhorst EL Flow, Bronkhorst IQ Flow, Brooks 5850). Each time, when one of the flow controller settings was changed, the flow rates were measured using a primary volumetric standard (Drycal, Definer 220), which has an accuracy of 0.75 % of the reading.

Synthetic air (79 % $N_2$ and 21 % $O_2$) was produced from evaporated high purity liquid $N_2$ and $O_2$ (Linde, purities > 99.9999 %). Impurities in the synthetic air supply are generally below the detection limits of analytical instruments (e.g., $CO < 10\,ppbv$, $NO + NO_2 < 10\,pptv$, hydrocarbons $< 50\,pptv$). Water vapour was produced by a controlled evaporation and mixing system (Bronkhorst, CEM) using pure water (Milli-Q). Ozone was produced by oxygen photolysis in synthetic air using the $185\,nm$ radiation from a low-pressure mercury lamp. In the flow tube, typical ozone mixing ratios were $35\,ppbv$ and the partial pressures of water vapour were in the range of 2.0 to $22.5\,hPa$.

The reactant gases CO, NO and $NO_2$ were supplied as certified mixtures in $N_2$ from commercial suppliers. The concentrations of the mixtures were controlled independently (Table 1). To measure CO concentrations, a small flow (cylinder A, Table 1) was diluted with a synthetic air flow, both controlled by mass flow controllers, and the resulting CO concentration was measured using a near-infrared cavity ring-down spectrometer (Picarro, G2401). This instrument has a high precision of a few ppbv and high linearity (Zellweger et al., 2012) and was calibrated against a CO standard from NPL (National Physical Laboratory, UK).

Two cylinders with different NO concentrations were used in this work (cylinders B and C, Table 1). For the analysis of the NO concentration in the cylinder B, a flow of the gas mixture was further diluted with $N_2$ using mass flow controllers. The resulting NO concentration was measured using a chemiluminescence instrument (Ecophysics, CLD770) for mixing ratios up to $100\,ppbv$. The instrument was calibrated using an NPL standard with a stated uncertainty of 0.8 %. The mixing ratios of NO



and $NO_2$ in the cylinders C and D (Table 1) were measured directly using a UV-VIS photometer (ABB, Limas 11HW) which is suitable for measurements up to $1,000\,\mathrm{ppmv}$. For all gas cylinders, the derived mixing ratios were found to be in agreement with the suppliers' specifications within the experimental uncertainties (Table 1). A weighted average of the measured and the supplier values was used to calculate the concentrations in the reaction kinetics experiments.

## 2.3   Kinetic analysis

The measured radical decay curves are expected to follow pseudo first order kinetics (Eq. 4, 5). The corresponding time-dependent OH fluorescence signals (photon counts $N(t)$) include a constant background signal, which is caused by scattered radiation from the probe laser and detector noise:

$$N(t) = N_0 \exp(-k't) + B \tag{6}$$

where $N_0$ is the initial fluorescence count and $B$ is the background.

The parameters $N_0$, $B$ and $k'$ were determined for the measured decay curves using a non-linear, least-square Levenberg-Marquardt fitting algorithm. The counts were weighted in the fit by their statistical errors, which follow Poisson statistics. The first $10\,\mathrm{ms}$ of the measured OH decay were generally discarded. The signal in this time period showed deviations from a single exponential behaviour and this time was necessary for the conversion of OH to $HO_2$ in the experiments with $HO_2$.

Experiments with zero air, which contained additionally only water vapour and ozone, were performed to determine the zero rate coefficient ($k_0$) of the OH and $HO_2$ decays caused by wall loss and potential gas-phase reactions in the zero gas (Eq. 7). For both radicals, the values were in the range of $(1.8 \pm 0.1)\,\mathrm{s^{-1}}$ for water vapour partial pressures between 2.0 and $22.5\,\mathrm{hPa}$.

The calculated, known contributions to the zero rate coefficient from gas-phase reactions were very small. The reaction of the added ozone (mixing ratio $35\,\mathrm{ppbv}$) with OH and $HO_2$ contributed only $0.06\,\mathrm{s^{-1}}$ and $0.0017\,\mathrm{s^{-1}}$, respectively, to the reactivity. The reactivity of self-reactions of OH and $HO_2$ radicals are also less than $0.07\,\mathrm{s^{-1}}$ and $0.04\,\mathrm{s^{-1}}$, respectively. The variability of the zero rate coefficient over the range of added water vapour concentrations gives an upper limit of $0.1\,\mathrm{s^{-1}}$ for the reactivity from potentially co-evaporated impurities of the water supply. The reactivity from potential impurities (e.g., CO, NOx, hydrocarbons) in the synthetic air supply can be estimated to have an upper limit of $0.1\,\mathrm{s^{-1}}$.

For these reasons, the zero rate coefficient was mainly determined by the lateral transport of radicals to the wall of the flow tube, where radicals are lost. This assumption is consistent with the diffusion of radicals, for which the mean quadratic displacement ($\langle \Delta r^2 \rangle$) can be calculated by Einstein's relation ($\langle \Delta r^2 \rangle = 4\,D\,t$, diffusion coefficients: $D(\mathrm{OH}) = 0.217\,\mathrm{cm^2 s^{-1}}$, $D(\mathrm{HO_2}) = 0.141\,\mathrm{cm^2 s^{-1}}$, Ivanov et al. (2007)).

In the experiments, the concentrations of the reactants were varied to determine the rate coefficients (Fig. 2). The reactant concentrations gave reactivities between 0 and $40\,\mathrm{s^{-1}}$. Approximately ten decay curves, each integrating 100 to 250 photolysis laser shots, were accumulated for each reactant concentration. The slope of a linear regression of the measured first order rate coefficients against the reactant concentration $[i]$ gives the second order reaction rate coefficient $k_{\mathrm{OH}+i}$. The intercept is the zero rate coefficient $k_0$:

$$k' = k_0 + k_{\mathrm{OH}+i}[i] \tag{7}$$





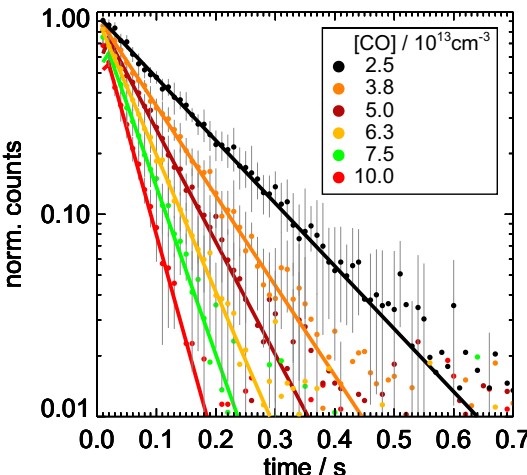

**Figure 2.** Examples of measured (dots) and fitted (lines) OH concentration decays (normalised to the fitted amplitude) for different CO concentrations measured at a temperature of $297\,\mathrm{K}$ and a pressure of $1\,\mathrm{atm}$ in this work. The background values determined by the fit are subtracted. For clarity, the measured decays are shown in the figure with a time resolution of $10\,\mathrm{ms}$. Error bars are 1-$\sigma$ statistical errors of the measurements.

## 3 Results and Discussion

### 3.1 Rate coefficient of the OH reaction with CO

The rate coefficient of the reaction of carbon monoxide (CO) with OH was studied in 4 experiments at room temperature (296 to $298\,\mathrm{K}$, Table A1), in each of which the OH reactivity was measured for 8 CO concentrations (Figure 3). As CO is transparent at a wavelength of $266\,\mathrm{nm}$ (Okabe, 1978), effects from the photolysis of CO in the flow tube by the photolysis laser can be excluded.

The experiments differed in the water vapour content with water vapour partial pressures between 3 and $20.5\,\mathrm{hPa}$. Since the rate coefficients agreed within 4 % and showed no trend with the presence of water vapour (Figure 3), a water vapour independent value of $k_{\mathrm{OH+CO}} = (2.38\pm0.11)\cdot10^{-13}\,\mathrm{cm^3\,s^{-1}}$ is determined from the weighted average of the rate coefficients determined at the different humidities (Table A1). The uncertainty is the total 2-$\sigma$ error, which is mainly due to the uncertainty in the CO concentration.

The reaction of OH with CO has been studied experimentally and theoretically over a wide range of conditions (e.g. Fulle et al., 1996; Atkinson et al., 2004; Johnson et al., 2014; Burkholder et al., 2020; Barker et al., 2020). It shows a complex non-Arrhenius temperature and pressure dependence. This can be explained by the formation of an activated radical intermediate, HOCO*, Smith and Zellner (1973)), which can be collisionally stabilised to HOCO or can decompose to $CO_2$ and an H atom (Reaction R1, R2).





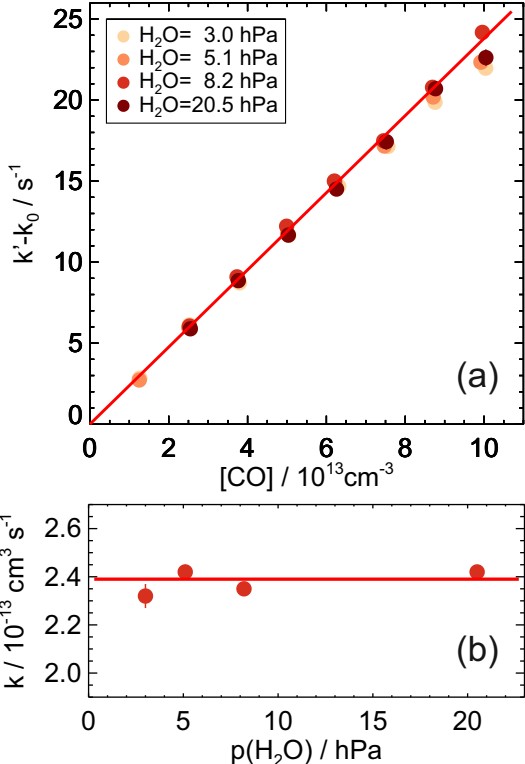

**Figure 3.** First order rate coefficients of the OH reaction with CO at room temperature $((297 \pm 1)$ K) and ambient pressure $((1017 \pm 8)$ hPa) in air and partial water vapour pressures between 3 and 20.5 hPa (upper panel). The zero rate coefficient $k_0$ is subtracted from the linear fit of the measured OH reactivity (Eq. 7). The slope of the red line is the weighted average of the second order rate coefficients determined at the different humidities. No dependence of the rate coefficient on water vapour is observed (lower panel). Error bars (1-$\sigma$ statistical errors) are partly smaller than the size of the symbols.

At high temperatures ($> 600$ K), HOCO becomes thermally unstable and forms OH and CO (Fulle et al., 1996), while in the atmosphere (200 to 300 K) it reacts mainly with $O_2$ to form $HO_2$:

$$HOCO + O_2 \rightarrow HO_2 + CO_2 \tag{R9}$$

The corresponding lifetime of HOCO is 130 ns in 1 atm pressure in air at a temperature of 298 K (Miyoshi et al., 1994).
Similarly, the H-atom produced in the decomposition of HOCO* (Reaction R2) reacts with $O_2$ to form $HO_2$ at a similar rate (Burkholder et al., 2020):

$$H + O_2 \rightarrow HO_2 \tag{R10}$$

Consistent with this mechanism, the OH decays measured in the present work showed a single exponential behaviour without regeneration of OH.



**Table 2.** Second order rate coefficients ($k$) of the OH reaction with CO measured absolutely in air or $N_2$ at ambient total pressure ($p$) and temperature ($T$). In addition, IUPAC and NASA-JPL recommended values are given for the conditions used in this work. Errors of the rate coefficients are 2-$\sigma$ uncertainties.

| $k/10^{-13}$ cm$^3$s$^{-1}$ | $T$/K | $p$/hPa | bath gas | $p(H_2O)$/hPa | reference |
|---|---|---|---|---|---|
| $2.18 \pm 0.50^a$ | 298 | 1013 | $N_2$ | 0.4–1.3 | Paraskevopoulos and Irwin (1984) |
| $2.30 \pm 0.11^b$ | 298 | 987 | $N_2$ | $< 0.2$ | Hofzumahaus and Stuhl (1984) |
| $2.35 \pm 0.20^a$ | 298 | 1013 | air | 0.013–27 | Hynes et al. (1986) |
| $2.44 \pm 0.37^a$ | 298 | 1013 | air | 0–27 | McCabe et al. (2001) |
| $2.29 \pm 0.28^c$ | 297 | 1017 | $N_2$ | - | IUPAC (Atkinson et al., 2006) |
| $2.43 \pm 0.12^c$ | 297 | 1017 | air | - | NASA-JPL (Burkholder et al., 2020) |
| $2.39 \pm 0.11^b$ | $297 \pm 1$ | $1017 \pm 8$ | air | 3.0–20.5 | this work |

$^a$linear fit of measured data to ambient conditions; $^b$measurement for stated conditions; $^c$parameterisation based on literature

Previous experimental studies at atmospheric temperatures have shown that the OH + CO reaction (Reaction R1) does not depend on the temperature at low pressure and shows only a small decrease (about 10 %) as the temperature increases from 200 to 300 K at a pressure of 1 atm (McCabe et al., 2001; Liu and Sander, 2015). The rate coefficient has a linear pressure dependence and increases by a factor of 1.6 in the pressure range from 0 to 1 atm (Atkinson et al., 2004; Burkholder et al., 2020).

Despite its importance in atmospheric chemistry, only two absolute measurements of the rate coefficient have been reported in previous studies at room temperature and 1 atm pressure of air (Table 2, Hynes et al. (1986); McCabe et al. (2001)). The data from these two studies are in very good agreement within 2 % with the results in this work. This is better than would be expected from the reported uncertainties of the pressure dependent expressions of the rate coefficient in Hynes et al. (1986) (12 %) and in McCabe et al. (2001) (15 %) and the measurement error of 5 % in this work.

Other studies have investigated the rate coefficient in $N_2$ at ambient pressure. The values agree within 5 to 10 % with measurements in air (Table 2). This is consistent with the experiments of Hynes et al. (1986) and McCabe et al. (2001), which show that the collisional stabilisation of the reactive complex is the same for $N_2$ and $O_2$ within the experimental uncertainties. However, experiments in $N_2$ require great care to avoid oxygen impurities, as H-atoms (Reaction R2) could react not only with molecular oxygen to form $HO_2$ (Reaction R10), but also with $HO_2$, thereby regenerating OH. This can lead to an apparent

reduction in the effective rate coefficient (Hofzumahaus and Stuhl, 1984; Paraskevopoulos and Irwin, 1984; Liu and Sander, 2015). Measurements in air, as in this work, avoid this potential problem because the oxygen concentration is high and any H-atoms react exclusively with $O_2$.

The results of this work are in good agreement with all previously reported absolute measurements in $N_2$ and air (Table 2) and are well within the uncertainties of recent recommendations from IUPAC (Atkinson et al., 2004) and NASA-JPL (Burkholder

et al., 2020). The IUPAC recommended value is only 5 % lower and the NASA-JPL recommended value is 3 % higher than





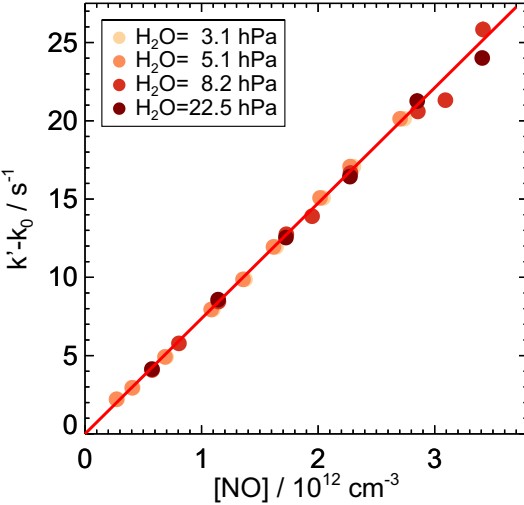

**Figure 4.** First order rate coefficients of the OH reaction with NO at ambient temperature $((297 \pm 1)\,\mathrm{K})$ and pressure $((1019 \pm 3)\,\mathrm{hPa})$ in air and various partial pressures of water vapour. The zero rate coefficient $k_0$ is subtracted from the linear fit of the measured OH reactivity (Eq. 7). The slope of the red line is the weighted average of the second order rate coefficients determined at the different humidities, as there is no observed dependence of the rate coefficient on water vapour. Error bars (1-$\sigma$ statistical errors) are partly smaller than the size of the symbols.

the value in this work (Fig. 3). The small discrepancies suggest that the uncertainties of the recommended values are likely overestimated by a factor of 2 at atmospheric pressure, although the uncertainty over the full range of the fall-off region may be higher.

No water vapour dependence of the rate coefficient was observed. The variability of the values $((3 \pm 3)\,\%)$ gives an upper limit
for the collisional stabilisation by water relative to air of 10. This agrees with previous measurements, where the efficiency was found to be a factor of 10 higher than that of $N_2$ at low pressures (up to $27\,\mathrm{hPa}$) in pure water and helium (Paraskevopoulos and Irwin, 1984). At atmospheric pressure, however, no significant effect of water vapour on the rate coefficient could be detected for partial water vapour pressures up to $27\,\mathrm{hPa}$ (McCabe et al., 2001). Based on these studies and results in this work, relevant water vapour effects due to clustering of water molecules with OH as assumed in previous experimental work (Beno et al.,
1985) or with HOCO as discussed in a theoretical study (Aloisio and Francisco, 2000) can be ruled out to be significant for atmospheric conditions at room temperature.

## 3.2 Rate coefficient of the OH reaction with NO

The rate coefficient of the OH reaction with NO (Reaction R3) was measured in air at a pressure of $1019\,\mathrm{hPa}$ and a temperature of $297\,\mathrm{K}$. In addition, the partial pressure of water vapour was varied between 3.1 and $22.5\,\mathrm{hPa}$. NO was provided by two gas
standards (Table 1).





**Table 3.** Second order rate coefficient ($k$) of the OH reaction with NO in air or $N_2$ at ambient total pressure ($p$) and temperature ($T$). In addition, values calculated from the parametrisations in the IUPAC and NASA-JPL recommendations and reported in the literature are given for the conditions in this work. Errors in the rate coefficients are 2-$\sigma$ uncertainties.

| $k/10^{-12}\,\mathrm{cm^3 s^{-1}}$ | $T/\mathrm{K}$ | $p/\mathrm{hPa}$ | bath gas | $p(\mathrm{H_2O})/\mathrm{hPa}$ | reference |
|---|---|---|---|---|---|
| $4.2 \pm 0.8^a$ | 298 | 1026 | $N_2$ | $< 4$ | Overend et al. (1976) |
| $6.7 \pm 3.3^b$ | 296 | 1013 | $N_2$ | 0.4 | Anastasi and Smith (1978) |
| $22 \pm 2^a$ | 295 | 985 | $N_2$ | - | Sharkey et al. (1994) |
| $7.4 \pm 1.3^c$ | 297 | 998 | $N_2$ | - | Bohn and Zetzsch (1997) |
| $7.1 \pm 0.4^d$ | 297 | 980 | $O_2$ | - | Bohn and Zetzsch (1999) |
| $6.3 \pm 0.5^a$ | 298 | 954 | $N_2$ | - | Sun et al. (2022) |
| $6.5 \pm 0.5^b$ | 297 | 1019 | $N_2$ | – | Sun et al. (2022) |
| $9.9 \pm 3.8^e$ | 297 | 1019 | $N_2$ | - | IUPAC (2017b) |
| $7.5 \pm 1.5^e$ | 297 | 1019 | air | - | NASA-JPL, Burkholder et al. (2020) |
| $7.3 \pm 0.4^a$ | $297 \pm 1$ | $1019 \pm 3$ | air | 3.1–22.5 | this work |

$^a$measurement for stated conditions; $^b$parameterisation based on measured data; $^c$derived from bi-exponential OH decays in a complex reaction system containing $H_2O_2$ and NO; $^d$derived from bi-exponential OH decays in a complex reaction system containing $H_2O_2$ and benzene; $^e$parameterisation based on literature

In the evaluation of the rate coefficient, systematic errors due to side reactions of the NO reactant need consideration. First, an influence of the photolysis laser on the NO concentration can be excluded, because NO does not absorb at 266 nm (Okabe, 1978). However, a small effect is expected from the reaction of NO with ozone forming $NO_2$ in the reaction volume. Under the experimental conditions in this work, a gradual decrease of NO by 1.6 % is expected in the flow tube before the air is sampled

by the inlet of the LIF detection cell, using a rate coefficient of the NO reaction with $O_3$ of $k_{\mathrm{NO+O_3}} = 1.9 \cdot 10^{-14}\,\mathrm{cm^3 s^{-1}}$ at a temperature of 298 K (Burkholder et al., 2020). As the rate coefficient for the reaction of OH with $NO_2$ is 1.6 times faster than that with NO, a small bias of $+1$ % can be estimated for the determination of the OH reaction rate with NO due to the formation of $NO_2$. Measured values (Table A1) are corrected for this bias.

The rate coefficient of the OH reaction with NO was derived from the slope of the measured OH reactivity when the NO

concentration was varied between 0.2 and $3.5 \cdot 10^{12}\,\mathrm{cm^{-3}}$ (Fig. 4, Table A1). The partial water vapour pressure was changed between 3.1 and 2.5 hPa. No significant effect of water vapour on the rate coefficient was observed. The weighted average of the slopes derived from measurements at 4 different water vapour concentrations gives a rate coefficient of $k_{\mathrm{OH+NO}} = (7.3 \pm 0.4) \cdot 10^{-12}\,\mathrm{cm^3 s^{-1}}$ at a pressure of $(1019 \pm 3)$ hPa and a temperature of $(297 \pm 1)$ K in air. The total uncertainty is mainly due to the uncertainty in the NO concentrations.

The differences between the rate coefficients in this work and the values recommended by NASA-JPL (Burkholder et al., 2020) and determined by Bohn and Zetzsch (1997) and Bohn and Zetzsch (1999) are less than 4 % (Tab. 3). A recent work by Sun et al. (2022) provides a parametrisation of the rate coefficient from measurements over a broad pressure range (15





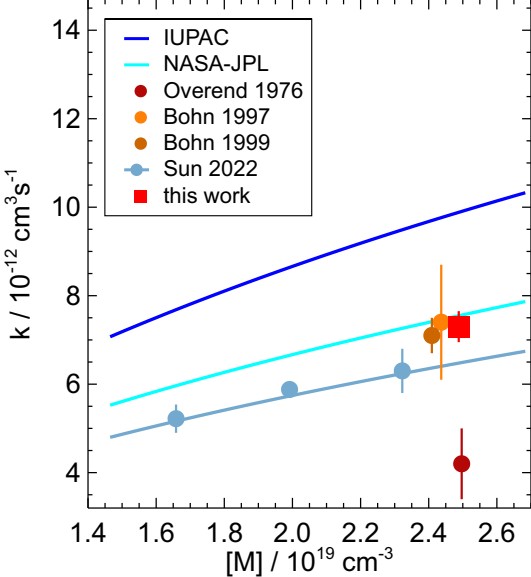

**Figure 5.** Pressure dependence of the second order rate coefficient of the OH reaction with NO reported in the literature. Data points represent measured values and solid lines represent parameterisations using Troe equations. The parameterisation of Sun et al. (2022) (line) is calculated for pure $N_2$ at 297 K, while the corresponding data points were measured at 298 K. Error bars are total errors. The high value measured by Sharkey et al. (1994) (Table 3) is not shown.

to 990 hPa) at different temperatures (273 K, 298 K, 333 K). Their parameterisation gives values which are approximately 13 % lower than those recommended by NASA-JPL for the experimental conditions in this work, but agrees better at lower

pressures. The measurements in Sun et al. (2022) were carried out in $N_2$. The authors assume that the collisional stabilisation of the activated association complex by $N_2$ and $O_2$ is similar so that their parameterisation can also be used for air.

The values recommended by IUPAC (2017b) are 35 % to 50 % higher than the measurements by Sun et al. (2022), Bohn and Zetzsch (1997, 1999) and this work (Figure 5), suggesting that the IUPAC recommendation may need to be revised.

Sun et al. (2022) also investigated the effect of water vapour on the rate coefficient. Measurements at various water concen-

trations at low pressure (66 hPa) and room temperature showed a rate coefficient at a water vapour partial pressure of 12 hPa that was 60 % higher than in pure $N_2$. The authors explained this behaviour by the more efficient collisional stabilisation of the activated association complex by water molecules, which was estimated to be a factor of 5 to 6 more efficient than that of $N_2$.

Sun et al. (2022) derived a Troe equation using different low-pressure rate coefficients for $N_2$ and water vapour following the approach described in Amedro et al. (2020). Using the values in Sun et al. (2022), the difference between the rate coefficients

for the lowest and highest water vapour concentrations in the experiments in this work is 5 % (Fig. 6). This is higher than the variability of the measured rate coefficients (1 %) determined at the different water vapour concentrations in this work and is a significant discrepancy.





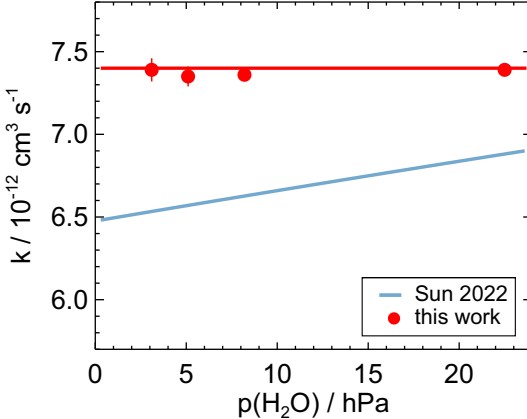

**Figure 6.** Water vapour dependence of the second order rate coefficient of the OH reaction with NO measured in this work and calculated from the parameterisation derived in Sun et al. (2022) for the conditions of this work ($T = 297\,\mathrm{K}$, $p = 1019\,\mathrm{hPa}$). The horizontal red line is the weighted average of the data measured in this work (Table A1). Error bars ($1\text{-}\sigma$ statistical errors) are partly smaller than the size of the symbols.

Liessmann et al. (2011) studied the influence of water on the reaction of OH with NO at low total pressures ($< 10\,\mathrm{hPa}$) and low temperatures (60 to $300\,\mathrm{K}$). They observed a strong enhancement of the rate coefficient of up to $40\,\%$ at a water vapour mixing ratio of $3\,\%$ at temperatures below $135\,\mathrm{K}$ in a Laval nozzle gas expansion, but the enhancement disappeared at room temperature and became hardly detectable.

The results of this work suggest that the efficiency of the collisional stabilisation by water at the conditions in the lower troposphere is smaller than predicted by the parameterisation in Sun et al. (2022). The different behaviour may be due to invalid assumptions in the determination of the parametrisation or due to undetected measurement errors in the data of Sun et al. (2022) or in the present work. More studies are required to resolve this discrepancy.

### 3.3  Rate coefficient of the OH reaction with $NO_2$

The rate coefficient of the OH reaction with $NO_2$ was measured at a pressure of $1034\,\mathrm{hPa}$, a temperature of $295\,\mathrm{K}$ and two water vapour partial pressures (6.2 and $17.6\,\mathrm{hPa}$) (Table A1). The values were determined from the slope of OH reactivity measurements with varying $NO_2$ concentrations.

The reaction of $NO_2$ with OH can produce either nitric acid ($HNO_3$, Reaction R4) or pernitrous acid (HOONO, Reaction R5). The ratio of the products, HOONO to $HNO_3$, increases with pressure and is $(14.2 \pm 1.2)\,\%$ for the experimental conditions of this work (Mollner et al., 2010). Pernitrous acid is thermally unstable and decomposes back to OH and $NO_2$. Its chemical lifetime is approximately $1.2\,\mathrm{s}$ at room temperature calculated using the NASA-JPL rate coefficients of the forward reaction and the equilibrium constant (Burkholder et al., 2020). This is a factor of 6 to 36 longer than the OH lifetimes in the experiments in this work. Therefore, the OH decays are expected to represent the sum of the two OH loss reaction channels



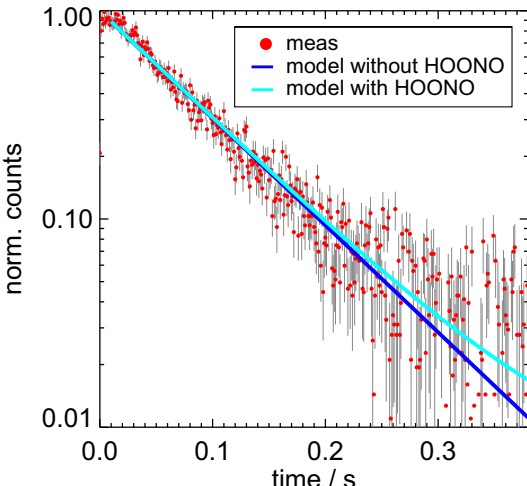

**Figure 7.** Example of the measured OH decay (normalised counts) and results of simulated decays including either only the OH loss in the reaction with $NO_2$ ("without HOOONO") or additionally the OH production from the HOONO decomposition ("with HOONO"). In the example, the corrected OH loss rate is $12.8\,s^{-1}$ and the measurement was performed with a $NO_2$ concentration of $1 \cdot 10^{12}\,cm^{-3}$ and a water vapour partial pressure of $6.2\,hPa$.

with little influence of the OH regeneration by the re-dissociation of HOONO. In agreement with the expectation, the observed OH decays showed no obvious deviation from a single exponential behaviour and were first fitted with the expression in Eq. 4.

In order to estimate the small effect of the HOONO decomposition on the derived rate coefficients, the OH decay curves were simulated for two cases, using a model that either included or excluded the HOONO decomposition (Fig. 7). The model

uses the value of the equilibrium constant (Reaction R5) by NASA-JPL ($K_{eq} = 2.2 \cdot 10^{12}\,cm^{-3}$, Burkholder et al. (2020)) and the branching ratio determined by Mollner et al. (2010). The results in Fig. 7 show that the two simulations agree well with the measured OH decay over the first order of magnitude and start to diverge from each other only after more than three OH lifetimes, where the noise of the measured decay curve becomes large.

The ratio of the two simulated OH decay curves was used to correct the measured OH decay for the OH production from

the HOONO decompositions. Fitting the corrected decay curves to a single exponential function (Eq. 4) gave 3 to 5 % higher decay rates than without the corrections. The largest effects are obtained for the lowest $NO_2$ concentrations. The corrected OH decay rates (Table A1) were used to calculate the rate coefficients of the OH reaction with $NO_2$ (Fig. 8).

Some other possible systematic errors in the determination of the rate coefficient can be ruled out. (1) The reaction of $NO_2$ with $O_3$ is far too slow ($k_{NO_2+O_3} = 3.2 \cdot 10^{-17}\,cm^3s^{-1}$, Burkholder et al. (2020)) to cause a significant change of the

$NO_2$ concentration in the flow tube at the given experimental conditions. (2) Although $NO_2$ absorbs at the wavelength of the photolysis laser (266 nm), the effect is negligible since less than $10^{-4}$ of the $NO_2$ molecules are photodissociated (absorption cross section $\sigma_{NO_2} = 2 \cdot 10^{-20}\,cm^2$, Vandaele et al. (1998)) at unity quantum yield. (3) Impurities of NO in the $NO_2$ gas mixture in the gas cylinder showed no detectable impurity of NO (Table 1) and therefore did not affect the determination of



**Table 4.** Second order rate coefficient ($k$) of the OH reaction with $NO_2$ in air or $N_2$ at ambient total pressure ($p$) and temperature ($T$). In addition, values calculated from the parameterisation in the IUPAC and NASA-JPL recommendations and reported in the literature are given for the conditions in this work. Errors in the rate coefficients are 2-$\sigma$ uncertainties.

| $k/10^{-11}\,\mathrm{cm^3s^{-1}}$ | $T/\mathrm{K}$ | $p/\mathrm{hPa}$ | bath gas | $p(\mathrm{H_2O})/\mathrm{hPa}$ | reference |
|---:|---:|---:|:---:|---:|:---|
| $1.40 \pm 0.1^a$ | 298 | 990 | air | 3.7 | Sadanaga et al. (2006) |
| $1.06 \pm 0.1^b$ | 298 | 1013 | air | - | Mollner et al. (2010) |
| $1.21 \pm 0.1^b$ | 298 | 1013 | air | - | Amedro et al. (2019) |
| $1.25 \pm 0.2^a$ | 293 | 1000 | $N_2$/air | - | Winiberg et al. (2020) |
| $1.08 \pm 0.1^b$ | 295 | 1034 | air | - | Mollner et al. (2010) |
| $1.26 \pm 0.1^b$ | 295 | 1034 | air | - | Amedro et al. (2019) |
| $1.22 \pm 0.15^b$ | 295 | 1034 | $N_2$/air | - | Winiberg et al. (2020), JPL-expression |
| $1.18 \pm 0.55^c$ | 295 | 1034 | $N_2$ | - | IUPAC (2017c) |
| $1.28 \pm 0.34^c$ | 295 | 1034 | air | - | NASA-JPL (Burkholder et al., 2020) |
| $1.23 \pm 0.04^{a,d}$ | 295 | 1034 | air | 6.2, 17.6 | this work |

$^a$ measurement for stated conditions; $^b$ parameterisation based on measured data; $^c$ parameterisation based on literature; $^d$ decay curves corrected for HOONO decomposition

the rate coefficient. (4) The formation of $NO_2$ dimers ($N_2O_4$) was insignificant as their estimated concentration was about

$1.4 \cdot 10^6\,\mathrm{cm^{-3}}$ at the maximum $NO_2$ concentration ($2.5 \cdot 10^{12}\,\mathrm{cm^{-3}}$) used in the experiments.

The type of bath gas may also affect the results, as the relative efficiency of the collisional stabilisation of the activated association complex by $O_2$ to $N_2$ is in the range of 0.67 (Mollner et al., 2010) and 0.74 (Amedro et al., 2019). As the experiments in this work were carried out in humidified synthetic air, the measured values refer to rate coefficients in a mixture of 79 % $N_2$ and 21 % $O_2$ and variable traces of water vapour.

The second order rate coefficients of the OH reaction with $NO_2$ obtained in this work obtained for water vapour partial pressures of 6.2 and 17.2 hPa differed by only 3.3 %, which is slightly higher than the combined statistical errors ($\pm 2.2$ %). The weighted average gives a value of $k_{\mathrm{OH+NO_2}} = (1.23 \pm 0.04) \cdot 10^{-11}\,\mathrm{cm^3s^{-1}}$. The total error includes the uncertainty of the reactant concentration.

The second order rate constant in air measured in this work is 4 % higher than the value recommended by IUPAC (2017c)

and is 4 % lower than the value recommended by NASA-JPL (Burkholder et al., 2020) (Table 4). The differences between the IUPAC and NASA-JPL recommendations become much larger at low pressure and low temperature as discussed in Amedro et al. (2019). The IUPAC recommended values are given for $N_2$ as a bath gas, whereas the NASA-JPL recommendation takes into account the differences in the collisional stabilisation of the activated association complex by $N_2$ and $O_2$. If this effect was taken into account in the IUPAC recommendation, the rate coefficient would be approximately 3 % lower (Amedro et al.,

2019), further increasing the difference between the values of the IUPAC and NASA-JPL recommendations. This also affects




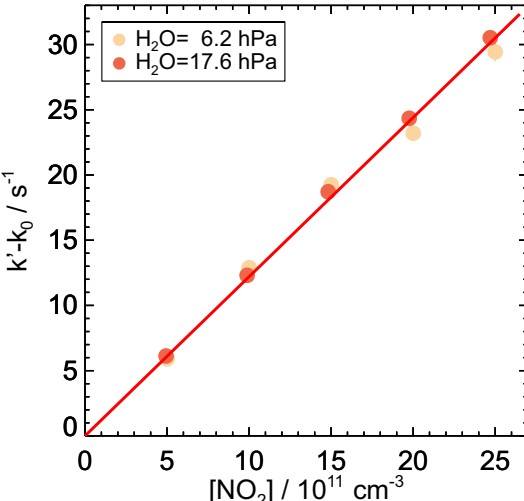

**Figure 8.** First order rate coefficients of the OH reaction with $NO_2$ from OH reactivity measurements at ambient temperature ($295\,\text{K}$) and pressure ($1034\,\text{hPa}$) in air at two partial pressures of water vapour. The zero rate coefficient $k_0$ is subtracted from the linear fit of the measured OH reactivity (Eq. 7). The slope of the red line is the weighted average of the second order rate coefficients determined at the different humidities, as there is no observed dependence of the rate coefficient on water vapour. Error bars ($1\text{-}\sigma$ statistical errors) are partly smaller than the size of the symbols.

the agreement with the value determined in this work. However, the resulting difference is still smaller than the uncertainty of the recommendations (Table 4).

Few other studies have measured the rate coefficient at ambient pressure, which is in the fall-off region (Fig. 9). The rate coefficients in two recent studies (Amedro et al., 2019; Winiberg et al., 2020) that derived Troe expressions (Eq. 1) agree well
to within $\pm 2\,\%$ (Table 4). The value obtained in the study by Mollner et al. (2010) is approximately $15\,\%$ lower than the values obtained in the more recent studies. Possible reasons for the lower value in Mollner et al. (2010) are discussed in Amedro et al. (2019), including possible systematic errors in the determination of the $NO_2$ concentration, but the exact reason remains unclear.

Amedro et al. (2020) determined the effect of collisional stabilisation of the association complex by water molecules in
experiments at low pressure and high partial pressures of water vapour. These experiments show that the collision efficiency is 6 times higher than for $N_2$, similar to the effect on the OH reaction with NO (Sun et al., 2022). Using the Troe equation determined by Amedro et al. (2020) for $N_2$-$H_2O$ mixtures, the rate coefficient increases by $2.7\,\%$ for water vapour partial pressures of 6.2 and $17.2\,\text{hPa}$ tested in this work (Fig. 10). The prediction is in good agreement with the increase in the rate coefficients of $(3.3 \pm 2.2)$ observed in this work (Fig. 10).

Sadanaga et al. (2006) found that the reaction rate coefficient decreases by $18\,\%$ when the partial pressure of water vapour is increased from 4 to $29\,\text{hPa}$. The experimental conditions were similar to those in this work in terms of bath gas, temperature, and pressure, and similar OH reactivity instruments were used. It is worth noting that the rate coefficients determined in the





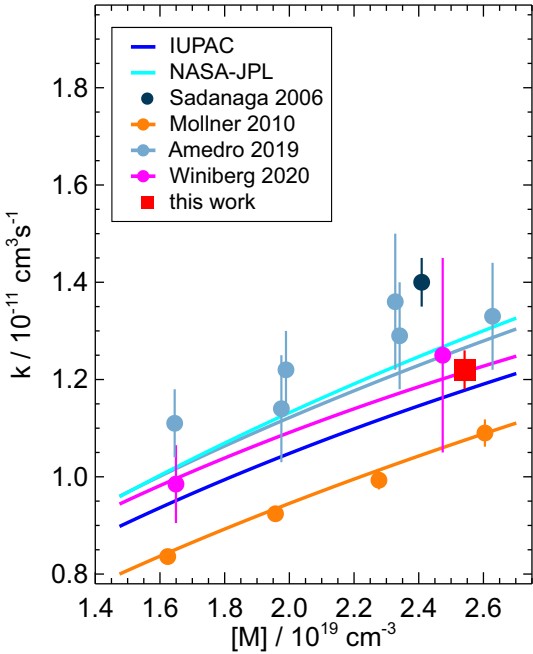

**Figure 9.** Pressure dependence of the second order rate coefficient for the OH reaction with $NO_2$ reported in the literature. Data points represent measured values and solid lines represent parameterisations using Troe equations. The parameterisation of Winiberg et al. (2020) is the NASA-JPL type Troe equation given in their work. The parameterisations are calculated for the temperature used in the present study (295 K), but the measured data points shown were obtained at slightly different conditions (Mollner et al. (2010): $T = 298$ K; Amedro et al. (2019): $T = 298$ K, $N_2$ bath gas; Winiberg et al. (2020): $T = 293$ K). Error bars are total errors.

work by Sadanaga et al. (2006) for water vapour partial pressures higher than 10 hPa are in good agreement with the rate coefficients in this work (Fig. 10). However, the increase of the rate coefficient at lower water vapour pressure contradicts the

results in the present work and in Amedro et al. (2020). Theoretical calculations in Sadanaga et al. (2006) could not explain their observed water vapour dependence. Therefore, the discrepancies with the recent studies remain unexplained.

### 3.4 Rate coefficient of the $HO_2$ reaction with $NO_2$

The reaction of $HO_2$ with $NO_2$ was studied in this work in air at a total pressure of 1031 hPa at a temperature of 297 K for different water vapour partial pressures between 2.0 hPa and 17.5 hPa. For these measurements, the instrument was operated

to produce $HO_2$ in the flow and to detect the $HO_2$ decay (Section 2.1). The $HO_2$ reaction with $NO_2$ forms pernitric acid ($HO_2NO_2$) in a termolecular reaction (Reaction R8).

Several potential systematic errors in the determination of the rate coefficients can be excluded:

- $HO_2NO_2$ is thermally unstable (Gierczak et al., 2005) and could affect the $HO_2$ decay by producing $HO_2$. The chemical lifetime of $HO_2NO_2$ was approximately 10 s calculated using the NASA-JPL equilibrium constant and reaction rates



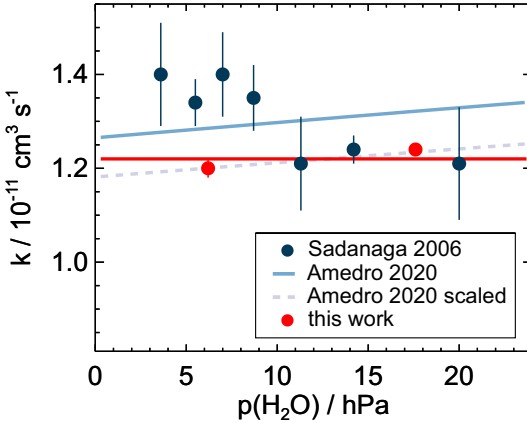

**Figure 10.** Water vapour dependence of the second order rate coefficient of the OH reaction with $NO_2$ measured in this work and calculated from the parameterisation of the measurements in Amedro et al. (2020) for conditions of this work (air, $T = 297\,\mathrm{K}$, $p = 1019\,\mathrm{hPa}$). The parameterisation fits the measured data of this work when scaled by a factor of 0.93. The measurements of Sadanaga et al. (2006) were made at at a temperature of 298 K and a pressure of 990 hPa in air.

(Burkholder et al., 2020). This is much longer than the timescale of the experiments in this work. Consequently, no deviations from a single-exponential behaviour were observed.

– As discussed for the OH reaction with $NO_2$ (Section 3.3), laser photolysis of $NO_2$ was negligible, since less than $10^{-4}$ of the $NO_2$ molecules were photolysed and thus the $NO_2$ concentration did not change.

– The reaction of $HO_2$ with NO from impurities in the $NO_2$ mixture of the air supply (Table 1) or from the $NO_2$ laser
photolysis could have contributed to the total $HO_2$ loss. However, the expected NO concentrations were very low and the NO reaction with $HO_2$ produces OH, which immediately reacted back to $HO_2$ in the reaction with excess CO (Section 2.1).

– Systematic errors due to the self-reaction of $HO_2$, as reported in previous studies (e.g. Kurylo and Ouellette, 1986; Christensen et al., 2004), were negligible due to the very low initial $HO_2$ concentrations used in this work (Section 2.3).

Measurements were performed at different water vapour partial pressures. The rate coefficients increased linearly by approximately 20 % as the water vapour partial pressure increased from 2.0 to 17.5 hPa (Fig. 11, Table A1). The observed linear dependence on water concentration can be empirically described by

$$k_{HO_2+NO_2}^{eff} = k_{HO_2+NO_2} + k_{HO_2+NO_2}^{H_2O}\,[H_2O] \tag{8}$$

where $k_{HO_2+NO_2}^{eff}$ is the measured second order rate coefficient determined from the observed $HO_2$ decays. The rate coefficient
$k_{HO_2+NO_2}$ represents the value in dry air at 1 atm and $k_{HO_2+NO_2}^{H_2O}$ is a third-order rate coefficient that describes the enhancement of the observed rate coefficient by water vapour. A linear fit of the measurements (Fig. 11) yields values of $k_{HO_2+NO_2} =$



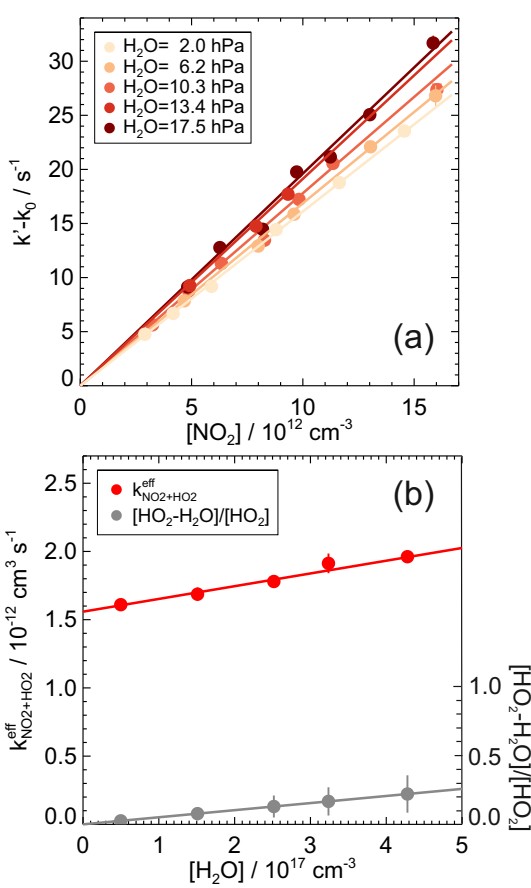

**Figure 11.** First order rate coefficients for the reaction of $HO_2$ with $NO_2$ at different humidities and ambient temperature (($297 \pm 1$) K) and pressure (($1026 \pm 5$) hPa) in air (upper panel). The zero rate coefficient $k_0$ is subtracted from the linear fit of the measured OH reactivity (Eq. 7). The lines are the results of a linear fit to the measurements at each humidity. The rate coefficients show a linear increase with the water vapour concentration, which scales with the concentration ratio of $HO_2$ complexed with $H_2O$ to free $HO_2$ radicals (lower panel). Error bars (1-$\sigma$ statistical errors) are partly smaller than the size of the symbols.





$(1.56 \pm 0.05) \cdot 10^{-12}\,\mathrm{cm^3 s^{-1}}$ and $k_{\mathrm{HO_2+NO_2}}^{\mathrm{H_2O}} = (0.92 \pm 0.09) \cdot 10^{-30}\,\mathrm{cm^6 s^{-1}}$. The errors are total uncertainties that include the measurement errors and the uncertainties in the $\mathrm{NO_2}$ and $\mathrm{H_2O}$ concentrations.

Sander and Peterson (1984) observed a similar behaviour and proposed an enhanced rate coefficient of the $\mathrm{NO_2}$ reaction with the hydrogen-bonded $\mathrm{HO_2 \cdot H_2O}$ complex:

$$\mathrm{HO_2 \cdot H_2O + NO_2 \rightarrow HO_2NO_2 + H_2O} \tag{R11}$$

The bimolecular rate coefficient of this reaction can be determined from the observed water vapour dependence, taking into account the chemical equilibrium between the free $\mathrm{HO_2}$ radical and the $\mathrm{HO_2 \cdot H_2O}$ complexes, which are in a fast equilibrium (e.g. Cox and Burrows, 1979; Aloisio et al., 2000):

$$\mathrm{HO_2 + H_2O \rightleftharpoons HO_2 \cdot H_2O} \tag{R12}$$

     Kanno et al. (2005) determined the value of the equilibrium constant at room temperature to be $K_{eq} = (5.2 \pm 3.2) \cdot 10^{-19}\,\mathrm{cm^3}$, which is in good agreement with the results of other studies (Cox and Burrows, 1979; Lii et al., 1981; Aloisio et al., 2000) and is also the value recommended by NASA-JPL (Burkholder et al., 2020), where the uncertainty is estimated to be a factor of two. The equilibrium can be assumed to be instantaneous on the time scale of the $\mathrm{HO_2}$ decay in the flow tube of the instrument

used in this work. The fraction $f$ of the free $\mathrm{HO_2}$ radicals can be estimated by the following approach:

$$f = \frac{[\mathrm{HO_2}]}{[\mathrm{HO_2}] + [\mathrm{HO_2 \cdot H_2O}]} = \frac{[\mathrm{HO_2}]}{[\mathrm{HO_2}] + K_{eq}[\mathrm{H_2O}][\mathrm{HO_2}]} = \frac{1}{1 + K_{eq}[\mathrm{H_2O}]} \approx 1 - K_{eq}[\mathrm{H_2O}] \tag{9}$$

where $K_{eq}[\mathrm{H_2O}]$ is a small number ($< 0.2$) for the range of water vapour concentrations used in this work and represents the concentration ratio of the complexed $\mathrm{HO_2}$ to free $\mathrm{HO_2}$ radicals (Fig. 11).

     The detection of $\mathrm{HO_2}$ in the low pressure detection cell of the instrument can be assumed to be equally sensitive to the

free $\mathrm{HO_2}$ radical and the $\mathrm{HO_2}$-water complex as indicated by routine $\mathrm{HO_2}$ measurements in the troposphere using the same detection method (e.g. Cho et al., 2023). Calibration measurements show that the $\mathrm{HO_2}$ detection sensitivity decreases slightly by 15 % with increasing water vapour concentrations in the range used in this work. This can be quantitatively explained by fluorescence quenching by water molecules (Fuchs et al., 2011), providing evidence for the same instrument sensitivity for the free $\mathrm{HO_2}$ radical and the $\mathrm{HO_2}$-water complex.

The observed radical decay using Eq. 9 is then given by:

$$
\begin{aligned}
\frac{d([\mathrm{HO_2}] + [\mathrm{HO_2 \cdot H_2O}])}{dt} &= (f \cdot k_{\mathrm{HO_2+NO_2}} + (1-f) \cdot k_{\mathrm{HO_2 \cdot H_2O + NO_2}})[\mathrm{NO_2}]([\mathrm{HO_2}] + [\mathrm{HO_2 \cdot H_2O}]) \\
&\approx (k_{\mathrm{HO_2+NO_2}} + (k_{\mathrm{HO_2 \cdot H_2O + NO_2}} - k_{\mathrm{HO_2+NO_2}}) \cdot K_{eq}[\mathrm{H_2O}]) \cdot [\mathrm{NO_2}]([\mathrm{HO_2}] + [\mathrm{HO_2 \cdot H_2O}]) \\
&= k_{\mathrm{HO_2+NO_2}}^{eff}[\mathrm{NO_2}]([\mathrm{HO_2}] + [\mathrm{HO_2 \cdot H_2O}])
\end{aligned}
\tag{10}
$$

This approach gives a linear dependence of the effective rate coefficient, $k_{\mathrm{HO_2+NO_2}}^{eff}$, on water vapour, as observed (Fig. 11):

$$k_{\mathrm{HO_2+NO_2}}^{eff} = k_{\mathrm{HO_2+NO_2}} + (k_{\mathrm{HO_2 \cdot H_2O + NO_2}} - k_{\mathrm{HO_2+NO_2}}) \cdot K_{eq}[\mathrm{H_2O}] \tag{11}$$





**Table 5.** Second order rate coefficients for the reaction of the free $HO_2$ radical and the $HO_2$-water complex with $NO_2$ and the third order rate coefficient describing the water dependence at ambient pressure ($p$) and temperature ($T$), and varying humidity, determined in this work. In addition to the IUPAC and NASA-JPL recommendations (values calculated for the experimental conditions), the results of experiments reported in the literature are given. The errors of the rate coefficients are 2-$\sigma$ uncertainties.

| reaction | rate coefficient | $T/K$ | $p/hPa$ | bath gas | $p(H_2O)/hPa$ | reference |
|---|---|---|---|---|---|---|
| $HO_2+NO_2$ | $(1.31 \pm 0.12) \cdot 10^{-12}\,cm^3s^{-1\ a}$ | 298 | 950 | $N_2$ | - | Bacak et al. (2011) |
| | $(0.76 \pm 0.19) \cdot 10^{-12}\,cm^3s^{-1\ b}$ | 297 | 1031 | $N_2$ | - | IUPAC (2017a) |
| | $(1.34 \pm 0.08) \cdot 10^{-12}\,cm^3s^{-1\ b}$ | 297 | 1031 | air | - | NASA-JPL (Burkholder et al., 2020) |
| | $(1.56 \pm 0.05) \cdot 10^{-12}\,cm^3s^{-1\ c}$ | 297 | 1031 | air | 0 | this work |
| $HO_2 \cdot H_2O+NO_2$ | $2.9 \cdot 10^{-12}\,cm^3s^{-1\ d,e}$ | 298 | 467 | $N_2$ | $0-21$ | Sander and Peterson (1984) |
| | $(3.4 \pm 1.1) \cdot 10^{-12}\,cm^3s^{-1\ d}$ | 297 | 1031 | air | $2.0-17.5$ | this work |
| $HO_2+NO_2+H_2O$ | $1.0 \cdot 10^{-30}\,cm^6s^{-1\ f}$ | 298 | 467 | $N_2$ | $0-21$ | Sander and Peterson (1984) |
| | $(0.92 \pm 0.09) \cdot 10^{-30}\,cm^6s^{-1\ f}$ | 297 | 1031 | air | $2.0-17.5$ | this work |

[a]measurement for stated conditions; [b]parameterisation based on literature; [c]calculated from the fit results Eq. 8 (Fig. 11); [d]calculated from the fit results Eq. 8 (Fig. 11) using $K_{eq} = 5.2 \cdot 10^{-19}\,cm^3$ (Kanno et al., 2005); [e]re-calculated; [f]termolecular reaction rate constant Eq. 8

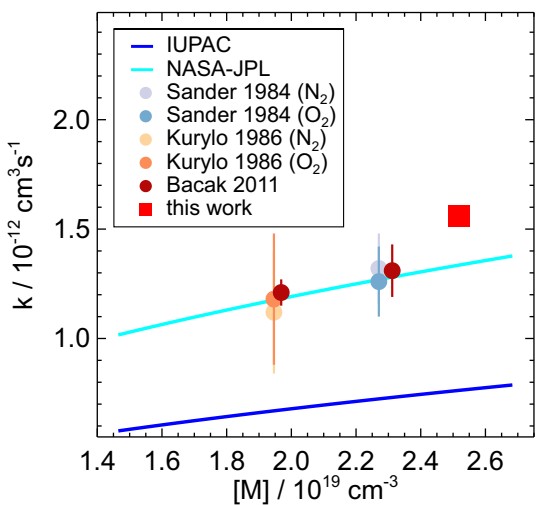

**Figure 12.** Pressure dependence of the second order rate coefficient of the $HO_2$ reaction with $NO_2$ reported in the literature and measured in this work. IUPAC and NASA-JPL values are calculated for the conditions of this work ($T = 297\,K$) and measurements of Bacak et al. (2011) were performed in $N_2$ at room temperature ($298\,K$) and a pressure of $933\,hPa$. Error bars are total errors.



Comparing the empirical expression of the $H_2O$ dependence (Eq. 8) with Eq. 11 and using the observed values (Table 5) allows to calculate the value of the second order rate coefficient for the $NO_2$ reaction with the $HO_2$-water complex (Reaction R11). This yields a value of $k_{HO_2 \cdot H_2O + NO_2} = (3.4 \pm 1.1) \cdot 10^{-12} \, \text{cm}^3 \text{s}^{-1}$. The uncertainty is higher than for the other rate coefficients due to the uncertainty of the equilibrium constant (Kanno et al., 2005).

The enhancement of the rate coefficient for the $HO_2$-water complex can be explained by the Chaperone mechanism, where the water molecule bonded to $HO_2$ acts as a third body that removes energy and stabilises the association product of the reaction between $HO_2$ and $NO_2$ similar to the mechanism discussed in Christensen et al. (2004) for the methanol bonded $HO_2$ radical.

To the best of our knowledge, our study provides the first experimental data on the $HO_2$ reaction with $NO_2$ in 1 atm air.
Previous studies were mostly carried out at lower total pressures and only the study by Bacak et al. (2011) was carried out under conditions close to those of this work (950 hPa in $N_2$ in a turbulent flow tube, Table 4, Fig. 12).

The rate coefficient derived in this work for the $NO_2$ reaction with the free $HO_2$ radical is a factor of 2 higher than the IUPAC (2017a) recommendation and 17 % higher than the NASA-JPL recommendation (Burkholder et al., 2020) (Table 5, Fig. 12). The NASA-JPL recommendation for room temperature is based on measurements by Sander and Peterson (1984);
Kurylo and Ouellette (1986); Christensen et al. (2004); Bacak et al. (2011). In these studies, the rate coefficients were measured at lower than ambient pressures between 250 and 950 hPa and temperatures between 277 and 298 K. A recent re-analysis of the rate coefficients available in the literature using a master equation analysis by McKee et al. (2022) gave a parametrisation, which agrees well with the values recommended by NASA-JPL.

The IUPAC recommendation is based only on measurements by Christensen et al. (2004), where experiments were per-
formed at much lower than ambient pressures ($< 270$ hPa). IUPAC excludes the studies by Sander and Peterson (1984) and Kurylo and Ouellette (1986) from their analysis because in these studies $HO_2$ was produced using methanol as a precursor. This can affect the results by the formation of a hydrogen-bonded adduct with $HO_2$, as the adduct can introduce a systematic error at temperatures below 250 K due to an increased rate of the $NO_2$ reaction with the methanol-$HO_2$ complex (Christensen et al., 2004). The large discrepancy between the NASA-JPL and IUPAC predictions in the fall-off region around 1 atm may
therefore be caused by the different data sets parameterised over different pressure ranges and/or by the use of different broadening factors in the Troe equations. Both recommendations underestimate the value determined in this work indicating the need for more extensive experimental studies covering a wider pressure range around 1 atm.

The effect of an increasing effective reaction rate in the presence of water was also observed in the experiments of Sander and Peterson (1984), which were carried out at room temperature and low pressure of 467 hPa with water vapour partial pressures
between 0 and 21 hPa in $N_2$. They determined a third order rate coefficient $k_{HO_2 + NO_2}^{H_2O}$ (Eq. 8) of $1.0 \cdot 10^{-30} \, \text{cm}^6 \text{s}^{-1}$, which is very close to the value of this work in 1 atm air (Table 5). It should be noted, however, that the expression for the rate coefficient $k_{HO_2 + NO_2}^{H_2O}$ includes the pressure dependent second order rate coefficient $k_{HO_2 + NO_2}$, so that the agreement is not necessarily expected. However, the pressure sensitivity is small at room temperature, because the value is mainly determined by the pressure independent rate coefficient $k_{HO_2 \cdot NO_2 + NO_2}$, which is a factor 2 to 3.5 higher than the pressure dependent rate
coefficient $k_{HO_2 + NO_2}$ for the conditions of the two studies.





Sander and Peterson (1984) also derived a relationship similar to that in Eq. 10 to determine the second order rate coefficient for the $NO_2$ reaction with the $HO_2$-water complex, but did not substitute $[HO_2]$ with $[HO_2] + [HO_2 \cdot H_2O]/(1 + K_{eq})$ in their rate equation. Using Eq. 11 and the latest recommendation for $K_{eq}$ (Kanno et al., 2005; Burkholder et al., 2020), the re-calculation of the second order rate coefficient gives a value of $2.9 \cdot 10^{-12} \, \mathrm{m^3 s^{-1}}$ for their data. The difference with the value in this work is 14 % but this is well within the measurement errors of both studies. The similarity of the values obtained in Sander and Peterson (1984) and in this work at different total pressures (467 hPa and 1031 hPa) supports the assumption that the reaction follows a Chaperone mechanism that is independent of the buffer gas ($N_2$, air).

An increase in the $HO_2$ reactivity due to the complexation with water molecules has been observed for other reactions, such as the self-reaction of $HO_2$, which can be enhanced by a factor of up to 2 in the moist troposphere (Lii et al., 1981; Kircher and Sander, 1984). Christensen et al. (2004) reported a similar effect for methanol, which also forms an adduct with $HO_2$ and increases the reaction rate between $HO_2$ and $NO_2$.

A temperature dependence of the rate coefficient of the $NO_2$ reaction with the $HO_2$-water complex can be estimated from measured data in Sander and Peterson (1984), which supports the assumption that the reaction of complexed $HO_2$ with $NO_2$ follows a bimolecular mechanism. In their work, they measured an increase in the value by a factor of 1.3 and 1.6, when the temperature was reduced from 298 K to 286 K and 275 K, respectively. Recalculation of their values (see above) gives a temperature trend with a positive Arrhenius activation energy $E/R = 1220$ K. In contrast, the pressure dependent $NO_2$ reaction with the free $HO_2$ radical shows a negative temperature dependence. The water dependence of the $HO_2$ reaction with $NO_2$ is therefore expected to increase in warmer regions. For example, increasing the temperature by 10 degrees at 298 K and 1 atm increases the ratio of the rate coefficients $k_{HO_2 \cdot H_2O + NO_2} : k_{HO_2 + NO_2}$ from 2.1 to 2.6.

Higher temperatures also mean higher concentrations of water vapour in the atmosphere. However, this does not necessarily increase the concentration of the $HO_2 \cdot H_2O$ complex, because the equilibrium (Reaction R12) is shifted towards free $HO_2$ radicals at higher temperatures. Overall, the influence of water vapour on the reaction of $HO_2$ with $NO_2$ is complex and remains uncertain, mainly because the equilibrium constant has a large uncertainty and because of the general lack of reaction kinetic measurements with water vapour over the tropospheric temperature range.

## 4   Conclusions

The second order rate coefficients of the termolecular reactions of OH with CO, NO and $NO_2$ and of the termolecular reaction of $HO_2$ with $NO_2$ were measured at tropospheric conditions of 1 atm pressure, room temperature and using humidified air as bath gas. The water vapour partial pressure was varied between 2.0 and 22.5 hPa. An instrument, which was developed for the measurement of atmospheric OH reactivity in field and chamber experiments (Lou et al., 2010), was used. This instrument measures the decay of OH radicals produced by laser flash photolysis of ozone using laser-induced fluorescence with a high sensitivity. The accuracies of the rate coefficients obtained in this work are better than 5 % mainly limited by the uncertainty of the certified commercial reactant gas standards, whose concentrations were checked using independent reference instruments.



Except for the rate coefficient of the $HO_2$ reaction with $NO_2$, the measured values are within the range of the recommendations of IUPAC and NASA-JPL evaluations, which partly specify large uncertainties. The experimental method used in this work yields rate coefficients which are among the most accurate values reported so far for atmospheric conditions. It is worth noting that the kinetic decays were carried out on a timescale similar to that of OH reactions in the lower troposphere. The initial OH concentrations were a factor of 10 to 10 000 lower than in all previous studies and the corresponding reactant concentrations were about a factor of 1000 lower, greatly reducing the potential for perturbation by secondary chemistry.

Measurements of the rate coefficient for the OH reaction with CO are in very good agreement with the NASA-JPL and IUPAC values. The differences are less than the 5 % uncertainty of the value determined in this work. The rate coefficient of the OH reaction with NO agrees within 3 % with the NASA-JPL value, whereas the IUPAC value is 35 % higher than in this work and 50 % higher than the recently measured value by Sun et al. (2022). This suggests that the IUPAC recommendation may need to be revised. The rate coefficient for the reaction of OH reaction with $NO_2$ measured in this work in air is 4 % lower than the NASA-JPL recommended value in air and is 4 % higher than the IUPAC recommended value. Since the collisional stabilisation of the activated association complex is different for $N_2$ and $O_2$ Amedro et al. (2019), the IUPAC value which is given for $N_2$ would be additionally 3 % lower in air.

Due to the large abundance of water vapour in the troposphere, it is an important question to what extent $H_2O$ influences atmospheric reactions by acting as a third collision partner or by forming a complex with OH or $HO_2$. In agreement with the literature, no significant influence of $H_2O$ was found for the reaction of OH with CO for water vapour partial pressures up to 22.5 hPa on water in 1 atm air at room temperature.

The activated association complexes formed in the OH reactions with NO and $NO_2$ have been shown to be better stabilised by water molecules than by $N_2$ and $O_2$ (Paraskevopoulos and Irwin, 1984; Amedro et al., 2020; Sun et al., 2022). However, the effect becomes small at pressures of 1 atm and water concentrations typically found in the lower troposphere and tested in this work.

For the reaction of OH with NO, a recent work by Sun et al. (2022) predicts an increase in the rate constant of up to 5 % for the range of water concentrations in this work. However, the observed variability in this work is only 1 % suggesting that the effect is smaller than expected from the results in Sun et al. (2022). The rate coefficients of the reaction of OH with $NO_2$ were measured for two water vapour partial pressures (6 hPa and 17 hPa). The small increase in the values of $(3.3 \pm 2.2)$ % with increasing water vapour agrees with a prediction of 2.7 % from the Troe equation determined by Amedro et al. (2020). A negative dependence on water vapour reported for atmospheric conditions in air by Sadanaga et al. (2006) could not be confirmed.

A strong water vapour dependence of the effective reaction rate coefficient was found for the reaction of $HO_2$ with $NO_2$ giving a second order rate coefficient for dry air at 1 atm pressure that is a factor of 2 larger than the recommendation by IUPAC for $N_2$ and 17 % higher than the NASA-JPL recommended value for air. The measured rate coefficient shows a linear increase by 25 % at a water vapour partial pressure of 17.5 hPa. Similar to the well-known water-dependent $HO_2$ self-reaction, the increased reactivity is presumably caused by $HO_2$ radicals that form a complex with water molecules, which reacts faster with $NO_2$ than free $HO_2$ radicals.



The observed increase can be explained by the known thermal equilibrium between free $HO_2$ radicals and $HO_2$ radicals complexed with $H_2O$ (Cox and Burrows, 1979; Aloisio et al., 2000). This can be used to determine the second order rate

coefficient for the reaction of the $HO_2 \cdot H_2O$ complex with $NO_2$ resulting in a value that is a factor of 2 faster than that of the reaction of free the $HO_2$ radical.

A re-analysis of the data of Sander and Peterson (1984), who studied the water vapour dependence of this reaction at a pressure of $467\,hPa$ and room temperature, gives a good agreement of the rate coefficients of the $HO_2 \cdot H_2O$ complex with $NO_2$ with the value determined in this work at $1031\,hPa$. This agreement supports the assumption that the reaction of the $HO_2 \cdot H_2O$

complex with $NO_2$ behaves like a pressure-independent bimolecular reaction. Although the rate coefficient of this Chaperone type reaction has a large uncertainty, the results suggest that the water effect should be included in atmospheric chemistry models. It also demonstrates the general need to consider potential water effects of reactions relevant in the atmosphere, as discussed in the review by Buszek et al. (2011) and shown in global chemical transport models (Khan et al., 2015).

Overall, the measurements in this work provide highly accurate rate coefficients that can serve as reference values at tropo-

spheric conditions and could be used to improve the parametrisation of termolecular rate coefficients (Burkholder et al., 2017; Fiore et al., 2024). The method of using an OH reactivity instrument for kinetic studies can be extended to also measure the temperature dependence of the rate coefficient, as successfully shown by Berg et al. (2024) for the OH reactions with alkanes, aromatics and monoterpenes.





## Appendix A: Measured rate coefficients

**Table A1.** Second order rate coefficients ($k$) determined in this study. Errors are 1-$\sigma$ statistical errors and do not include the uncertainty of the reactant's concentration.

| Reaction | $k\,/\,\mathrm{cm^3 s^{-1}}$ | $T\,/\,\mathrm{K}$ | $p\,/\,\mathrm{hPa}$ | $p(\mathrm{H_2O})\,/\,\mathrm{hPa}$ |
|---|---|---|---|---|
| OH + CO | $(2.32 \pm 0.05) \cdot 10^{-13}$ | 295 | 1009 | 3.0 |
| | $(2.42 \pm 0.03) \cdot 10^{-13}$ | 295 | 1009 | 5.1 |
| | $(2.35 \pm 0.01) \cdot 10^{-13}$ | 298 | 1022 | 8.2 |
| | $(2.42 \pm 0.01) \cdot 10^{-13}$ | 299 | 1025 | 20.5 |
| OH + NO | $(7.39 \pm 0.07) \cdot 10^{-12a}$ | 296 | 1016 | 3.1 |
| | $(7.28 \pm 0.06) \cdot 10^{-12a}$ | 296 | 1018 | 5.1 |
| | $(7.29 \pm 0.03) \cdot 10^{-12b}$ | 298 | 1023 | 8.2 |
| | $(7.32 \pm 0.03) \cdot 10^{-12b}$ | 298 | 1022 | 22.5 |
| OH + NO$_2$ | $(1.20 \pm 0.02) \cdot 10^{-11c}$ | 295 | 1034 | 6.2 |
| | $(1.24 \pm 0.01) \cdot 10^{-11c}$ | 295 | 1034 | 17.6 |
| HO$_2$ + NO$_2$ | $(1.61 \pm 0.02) \cdot 10^{-12}$ | 297 | 1021 | 2.0 |
| | $(1.69 \pm 0.05) \cdot 10^{-12}$ | 297 | 1031 | 6.2 |
| | $(1.78 \pm 0.02) \cdot 10^{-12}$ | 297 | 1031 | 10.3 |
| | $(1.73 \pm 0.04) \cdot 10^{-12}$ | 297 | 1032 | 13.4 |
| | $(1.96 \pm 0.02) \cdot 10^{-12}$ | 297 | 1031 | 17.5 |

[a]using the NO gas mixture from cylinder B (Table 1); [b]using the NO gas mixture from cylinder B (Table 1) [c]decay curves corrected for HOONO decomposition

*Data availability.* The data is listed in the Table in the Appendix.

*Author contributions.* HF and AH wrote the manuscript, MR performed the measurements and evaluated them to determine the rate coeffi-cients, AN and CE supported the measurements. All co-authors discussed the results.

*Competing interests.* At least one of the (co-)authors is a member of the editorial board of Atmospheric Chemistry and Physics. There are no other conflicts to declare.

*Acknowledgements.* We thank Christian Ehlers and Danielle Barnett for the support of the measurements of concentrations of the gas standards used in this work.



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
