# Peer review of "Kinetics of the reactions of OH with CO, NO, NO2 and of HO2 with NO2 in air at 1 atm pressure, room temperature and tropospheric water vapour concentrations"

_EGUsphere, 2024_

## Referee Comment (RC1)

The current manuscript describes the measurement of rate constants for some key reactions under real atmospheric conditions. The rate reactions of OH radicals with CO, NO and $NO_2$ as well as the reaction of $HO_2$ with $NO_2$ are measured at different relative humidity. The results for the 3 reactions of the OH radical agree well with recommendations, and will improve the uncertainty of the recommended rate constants. The rate for the reaction of $HO_2$ with $NO_2$ is faster than recommendations and shows strong dependence on RH: the parametrization should be considered in atmospheric chemistry models.

The experiments are carried out very carefully and the paper is very well written. A detailed description of the experimental conditions and data evaluation procedures is given and the comparison with the literature is comprehensive.

I strongly recommend publication of this manuscript, and have only some minor comments:

Line 18: remove ) from "in which $HO_2$) "

Line 97: for completeness you might want to cite a very recent paper describing a new FAGE-type instrument for measuring OH and $HO_2$ kinetics at high pressures: "New Instrument for Time-Resolved OH and $HO_2$ Quantification in High-Pressure Laboratory Kinetics Studies, Leonid Sheps and Kendrew Au, JPC A (2024), doi:10.1021/acs.jpca.4c00994"

You might also want to cite the following paper which uses a FAGE instrument to study the water dependence of a rate constant under atmospheric conditions, similar to the current work: "Water Vapor does not Catalyze the Reaction between Methanol and OH Radicals, W. Chao et al., Angewandte Chemie International Edition (2019), doi:10.1002/anie.201900711"

Line 146: when you give the number fore the detection limit ($1e6$ $cm^{-3}$) in 1 minute, is this the detection limit for the pulsed generation, which you use in this paper, i.e. the detection limit after averaging over 60 laser pulses or is this the detection limit for continuous OH generation?

Line 210: Concerning the zero-air loss of both radicals, it is surprising that both radicals give the same loss rate. It would be expected that the $HO_2$ loss is slower due to lower reactivity and also lower diffusion compared to OH. Any idea what might be the reason?

Line 296: 2.5 hPa is probably 22.5 hPa?

Line 384: (3.3+/-2.2) %, I imagine?

Line 435: Even though the absolute calibration of $HO_2$ is not really important for this work, I wonder how sure you are that the FAGE is equally sensitive to the complexed and the free $HO_2$, because there is no clear information in the cited reference Cho et al 2023.